# LOCUS (LOng Covid–Understanding Symptoms, events and use of services in Portugal): A three-component study protocol

J. P. Dinis Teixeira[1]*, Mário J. D. S. Santos[2], Patrícia Soares[2], Luísa de Azevedo[3], Patrícia Barbosa[1], Andreia Vilas Boas[4,5], João V. Cordeiro[2,6], Sónia Dias[2], Marta Fonseca[7], Ana Rita Goes[2], Maria João Lobão[2,8], Marta Moniz[2], Sofia Nóbrega[9], André Peralta-Santos[2], Víctor Ramos[1], João Victor Rocha[2], António Carlos da Silva[10], Maria da Luz Brazão[9], Andreia Leite[2,11], Carla Nunes[2]†

1 NOVA National School of Public Health, Universidade NOVA de Lisboa, Lisbon, Portugal, 2 NOVA National School of Public Health, Public Health Research Centre, Comprehensive Health Research Center, CHRC, NOVA University Lisbon, Lisbon, Portugal, 3 Serviço de Medicina Interna, Centro Hospitalar Póvoa de Varzim-Vila do Conde, EPE, Póvoa de Varzim, Portugal, 4 Hospital da Luz Arrábida, Vila Nova de Gaia, Portugal, 5 Portuguese Society of Internal Medicine, Lisbon, Portugal, 6 Centro Interdisciplinar de Ciências Sociais, Lisbon, Portugal, 7 NOVA Medical School, Comprehensive Health Research Centre, Universidade NOVA de Lisboa, Lisbon, Portugal, 8 Internal Medicine Department, Hospital de Cascais, Alcabideche, Cascais, Portugal, 9 Hospital Central do Funchal, Serviço de Saúde da Região Autónoma da Madeira, SESARAM, EPE, Funchal, Portugal, 10 Public Health Department, Regional Health Administration of Lisbon and Tagus Valley, Ministry of Health, Lisbon, Portugal, 11 Department of Epidemiology, Instituto Nacional de Saúde Doutor Ricardo Jorge, Lisbon, Portugal

☯ These authors contributed equally to this work.
† Deceased.
* jpd.teixeira@ensp.unl.pt

**Data Availability Statement:** No datasets were generated or analysed during the current study. All

## Abstract

Approximately 10% of patients experience symptoms of Post COVID-19 Condition (PCC) after a SARS-CoV-2 infection. Akin acute COVID-19, PCC may impact a multitude of organs and systems, such as the cardiovascular, respiratory, musculoskeletal, and neurological systems. The frequency and associated risk factors of PCC are still unclear among both community and hospital settings in individuals with a history of COVID-19. The LOCUS study was designed to clarify the PCC's burden and associated risk factors. LOCUS is a multi-component study that encompasses three complementary building blocks. The "Cardiovascular and respiratory events following COVID-19" component is set to estimate the incidence of cardiovascular and respiratory events after COVID-19 in eight Portuguese hospitals via electronic health records consultation. The "Physical and mental symptoms following COVID-19" component aims to address the community prevalence of self-reported PCC symptoms through a questionnaire-based approach. Finally, the "Treating and living with Post COVID-19 Condition" component will employ semi-structured interviews and focus groups to characterise reported experiences of using or working in healthcare and community services for the treatment of PCC symptoms. This multi-component study represents an innovative approach to exploring the health consequences of PCC. Its results are expected to provide a key contribution to the optimisation of healthcare services design.

relevant data from this study will be made available upon study completion.

**Funding:** This study is sponsored by Pfizer (grant code #68639655; URL: https://www.pfizer.pt/). The funders had and will not have a role in study design, data collection and analysis, decision to publish, or preparation of the manuscript.

**Competing interests:** The authors have declared that no competing interests exist.

## Introduction

In March 2020 the World Health Organization (WHO) declared COVID-19 a pandemic, affecting virtually all countries worldwide. After two years, 630 million cases and 6.5 million deaths have been registered [1], remaining a public health emergency of international concern [2].

Early in the pandemic, reports have emerged worldwide of persistent or *de novo* syndromes in patients after they recovered from their COVID-19 acute illness [3–6]. It is estimated that at least 10% of people infected with COVID-19 have persistent symptoms over the subsequent weeks after the acute phase [2,7–9]. Notwithstanding, the incidence of this condition varies widely by type of symptoms, age groups, severity of the original episode and between studies with different methodological approaches [10–13].

Given the heterogeneity in defining and naming this condition, WHO and several stakeholders reached a clinical definition describing Post COVID-19 Condition (PCC) in October 2021 [14]. PCC was defined as a *"condition [that] occurs in individuals with a history of probable or confirmed SARS-CoV-2 infection, usually three months from the onset of COVID-19 with symptoms that last for at least two months and cannot be explained by an alternative diagnosis. (. . .) Symptoms may be new onset following initial recovery from an acute COVID-19 episode or persist from the initial illness. Symptoms may also fluctuate or relapse over time."*.

Evidence so far has shown that female sex, older age, comorbidities, the severity of acute disease and obesity are important risk factors for the development of PCC [11,15,16]. However, further clarification is needed to better understand the risk and protective factors associated with PCC. Patients have reported a multitude of symptoms that may constitute a PCC diagnosis. The symptoms affected multiple organs and systems inter alia, cardiovascular (e.g., palpitations), neurological (e.g., cognitive symptoms, depression), respiratory (e.g., cough, shortness of breath) [3,17–21]. Other less commonly reported symptoms include anaphylaxis and new allergies, seizures, loss of vision and hearing, facial paralysis, and cutaneous signs [21–23]. Furthermore, these symptoms can occur, relapse and remittee [24,25].

Severe cardiovascular and respiratory complications of COVID-19 and PCC have also been the aim of extensive study in the last two years. Results have consistently pointed towards an increased incidence of cardiac injury—particularly myocardial -, thrombosis in multiple organ systems and respiratory function abnormalities in patients diagnosed with COVID-19 [26–32].

To date, PCC has mainly been explored from a medical standpoint, often overlooking the subjective impact of social and psychological aspects on health-related outcomes. Qualitative studies have shown PCC is perceived as difficult to control and poorly understood by patients [33,34]. The ability to capture this type of data makes qualitative research essential to better understand the phenomenon and to optimize healthcare services design.

Countries have put up a great effort to withstand the threat COVID-19 posed to the resilience of health systems worldwide [35,36]. Many of those systems were known to lack appropriate levels of preparedness before the COVID-19 pandemic [37] and have become particularly vulnerable to the supplementary stress that the PCC may represent in the near future. Research on PCC is crucial to identify how to best prepare health systems to address its impact [17]. Identifying risk factors and analysing patients' experiences may also help optimise decision-making processes in clinical practice.

### National context and study objectives

Because of its potential magnitude, furthering the knowledge on the incidence of PCC, its risk factors, and understanding the health professional and patients' experiences has become a

**Fig 1. Conceptualization of the three components of the study.**

pressing matter. Portugal had registered over six million COVID-19 cases by early August 2022 [38,39], which is in line with other high-income countries around the world in terms of cases per million [1]. The present study aims to characterise PCC with three distinct components designed to assess specific aspects and groups of patients in Portugal (Fig 1).

The study's objectives, by component, are: i) to estimate the incidence of cardiovascular and respiratory events after COVID-19, compare this incidence to the background rate and identify risk factors of cardiovascular and respiratory events in individuals hospitalised due to COVID-19 in Portugal; ii) to estimate the nine and 12-month prevalence of physical and mental symptoms, health-related quality of life and the impact of these symptoms on work and daily-life activities following COVID-19, compare them with non-COVID-19 patients, and identify symptoms' persistence risk factors; and iii) to understand the subjective experience of patients with PCC and healthcare professionals and explore the specific healthcare needs of these patients.

## Materials and methods

### Study design

The "Cardiovascular and respiratory events following COVID-19" component of the study will be based on data manually extracted from electronic medical records and will consist of a retrospective cohort that will include individuals hospitalised due to COVID-19 between

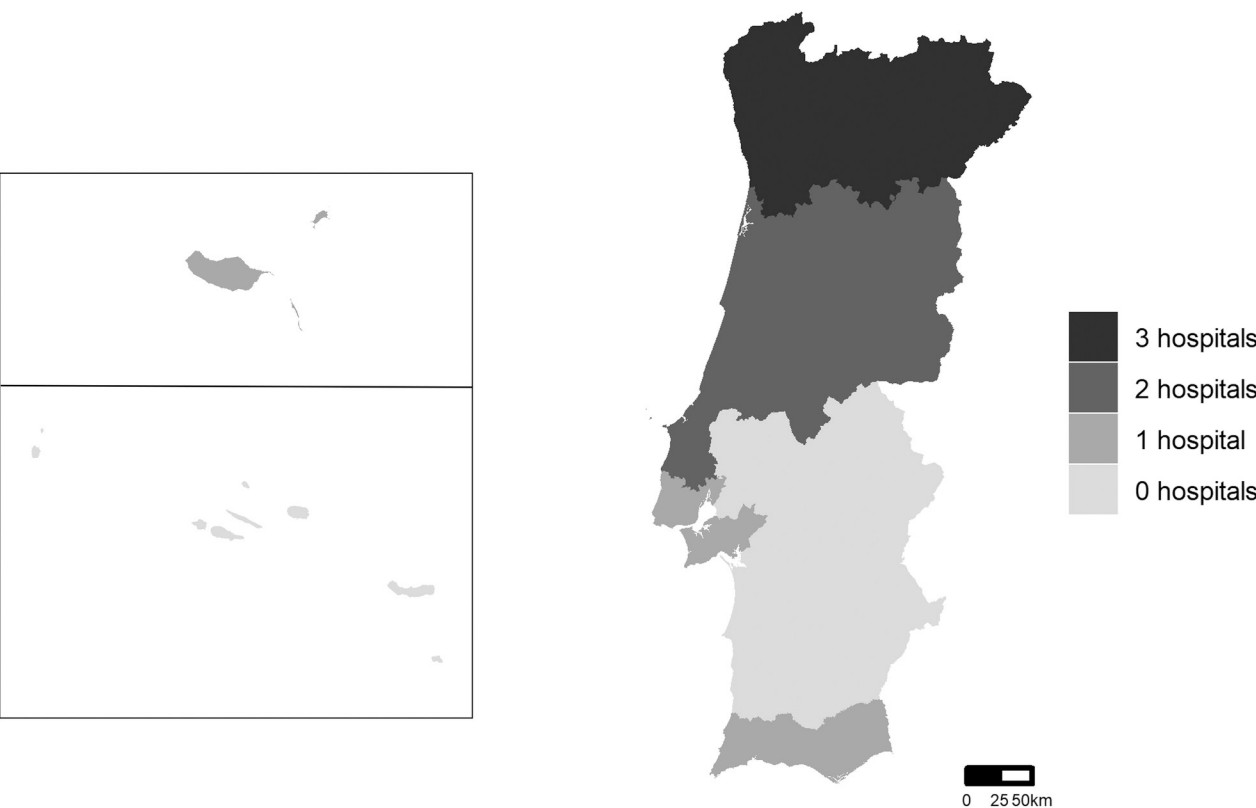

**Fig 2. Map depicting the geographic distribution of the hospitals that will be enrolled in the study: Algarve University Hospital Centre—Portimão Unit, Braga Hospital, Cascais Hospital, Coimbra University Hospital Centre, Funchal Central Hospital, Póvoa de Varzim / Vila do Conde Hospital Centre, Tondela-Viseu Hospital, Vila Real Hospital (Created with mapchart.net).**

March 2020 and March 2021 in the hospitals whose locations are depicted in Fig 2. We will exclude patients: below the age of 18 years old; who died during hospitalisation, with COVID-19 who were admitted due to another cause (e.g., pregnant women, emergency surgery), and individuals who do not live in Portugal. Patients will be identified in each hospital database by ICD-10 codes for COVID-19 attributed to each admission episode during the defined recruitment period.

Data (outcome and relevant patient and infection characteristics) will be collected following consultation of clinical files. Only the medical staff will have access to the clinical files and extract the information needed. Furthermore, in hospitals with a post-COVID clinic (namely, Hospital Central do Funchal), patients´ sociodemographic information, PCC clinical presentation and recovery will be characterised.

The "Physical and mental symptoms following COVID-19" component will be a questionnaire-based ambidirectional cohort with two groups of individuals: one with a positive and one with a negative real-time reverse transcriptase-polymerase chain reaction (RT-PCR) test result for SARS-CoV-2 nine months before the start of data collection. This component has two timepoints, the first nine months after the RT-PCR test and the second twelve months after. The participants selection will be based on medical notifications from the National Epidemiological Surveillance System (SINAVE). Based on data from SINAVE, the data owner (General Directorate of Health–DGS) will provide the research team with information on a list of potential participants in two phases (Fig 3).

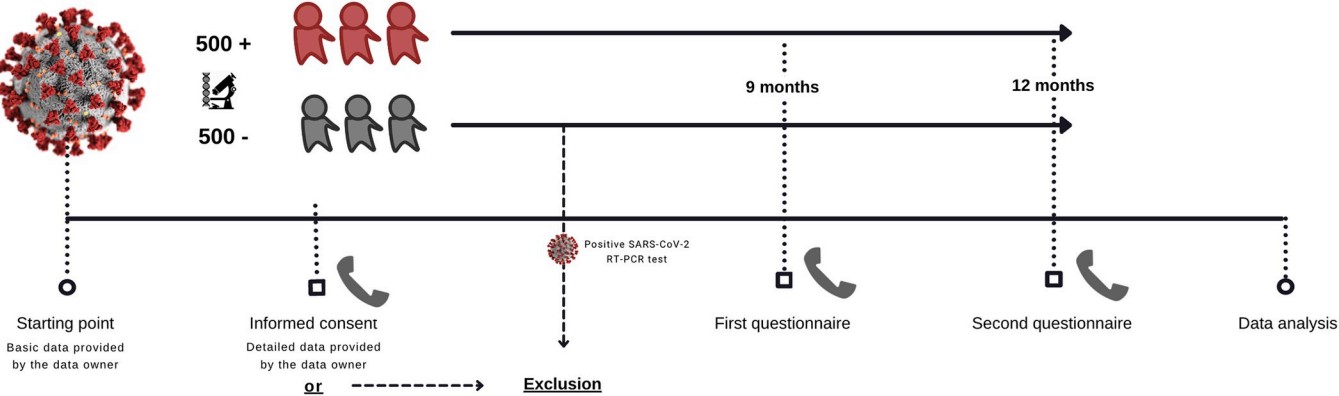

**Fig 3. Timeline with key points of the "Physical and mental symptoms following COVID-19" component data collection.**

In the first phase, the data owner will provide the names and contact numbers (landline or mobile) of individuals who had an RT-PCR test nine months before the beginning of data collection. Verbal informed consent will be obtained over the telephone by a trained inquirer. The full consent script will be read to the participant, where the participant can ask questions, refuse or accept to participate in the study. At this stage, the inquirer will have no knowledge of whether the participant had a positive or negative test. Informed consent will be applied by a small number of inquirers (planned to be two), with no further access by the remaining team (apart from the principal investigator), thus limiting the number of individuals team members accessing contacts of individuals. After the informed consent application, and for individuals who consented to participate in the study, the data owner will send us the birthdate and the result of the RT-PCR test for SARS-CoV-2. Inquirers will receive specific training to carry out this task and are bound by deontological professional secrecy rules. Calls may be scheduled at times that are most convenient for the participants.

We will exclude individuals: i) below the age of 18 years old; ii) who do not consent to participate; iii) without a valid landline or mobile phone registered; iv) institutionalised (e.g. residential structures for the elderly; prisons); v) who died between the date of the test and the call; vi) with language difficulties (languages not covered by the group of translators that are part of the team of investigators) or deafness, as well as advanced states of mental illness or dementia; vii) Portuguese tourists or emigrants on holidays in Portugal; and vii) those in the COVID-19 acute phase during the questionnaire application.

We aim to include 1000 participants (500 positive cases and 500 negative) at each time-point. This would allow us to identify an odds ratio of 2.1, assuming a prevalence of symptoms of 10% among positive cases, a power of 80% and a significance level of 5%. Assuming a participation rate of 80% and a loss to follow-up of 30%, we will invite 1900 individuals in each group at the first data collection point. In case a lower prevalence is estimated, power calculations will be performed to identify the increases in risk we would have been able to detect.

The "Treating and living with Post COVID-19 Condition" component of the study will have a qualitative research design, encompassing at least three focus group discussions with healthcare professionals and at least 24 semi-structured interviews with PCC patients. Professionals will be recruited for the focus groups through a snowball sampling strategy, starting from the contact network of the research team. Patients with PCC will be recruited from the group of participants of the "Physical and mental symptoms following COVID-19" component of the study who declared their availability for participating in subsequent stages of this research and consented to being contacted a second time by the research team. Participants'

age, gender, region and, in the case of PCC patients, educational level and PCC symptoms will be used as diversification criteria to promote the inclusion of different profiles of healthcare professionals and users, and to widen the diversity within each focus group discussion.

Although several studies have tried to get a precise assessment of PCC, many did not use adequate control groups nor have approached the entity from different perspectives at once, as the present study does.

## Ethics

The study design and protocol are compliant with fundamental research ethics norms and documents and have been reviewed and approved by the competent Ethical Review Boards. The first component of the study "Cardiovascular and respiratory events following COVID-19" was approved by the Ethics Committee for Health from each participanting hospital from each participating hospital (S2 Table). All hospitals waived the requirement for informed consent, as data will be completely anonymised. The second and third parts of the study, "Physical and mental symptoms following COVID-19" and "Treating and living with Post COVID-19 condition" were approved by the Ethics Committee for Health of the Regional Health Administration of Lisbon and Tagus Valley.

## Outcomes

In the "Cardiovascular and respiratory events following COVID-19" component of the study, our outcomes of interest are cardiovascular and respiratory events. Cardiovascular events will be defined as any of the following events after a COVID-19 diagnosis: heart failure, myocarditis, arrhythmia, acute myocardial infarction, pulmonary thromboembolism, intracardiac thrombus and deep vein thrombosis. Respiratory events will be defined as any of the following events after a COVID-19 diagnosis: pulmonary fibrosis, respiratory failure and obstructive or restrictive pulmonary disease.

In the "Physical and mental symptoms following COVID-19" component of the study, the outcome of interest is the existence of the symptoms stated in Table 1. Physical symptoms will be self-reported through a questionnaire based on the International Severe Acute Respiratory and emerging Infection Consortium (ISARIC) / WHO COVID-19 clinical characterisation protocol [40]. Mental symptoms will be measured using the Patient Health Questionnaire (PHQ)-2 to screen for depression, the General Anxiety Disorder (GAD)-2 questionnaire to

**Table 1. Symptoms whose presence will be assessed on the "Physical and mental symptoms following COVID-19" component.**

| | | |
|---|---|---|
| abdominal pain | altered bowel movements | altered chewing |
| altered feeling in one side of the body or face | altered menstrual bleeding | altered swallowing |
| altered urinary elimination | anorexia | Anxiety |
| balance problems | chest pain | Cough |
| depression | diarrhea | dysgeusia |
| dysosmia | dyspnoea | erectile dysfunction |
| fainting | fatigue | Fever |
| headache | heart palpitations | Insomnia |
| joint pain | lack of concentration | lack of memory |
| myalgia | nausea | odynophagia |
| post-traumatic stress disorder | rhinorrhea | Seizures |
| skin rash | tingling feeling | tinnitus and tremor |

screen for anxiety and The Primary Care PTSD Screen for DSM-5 (PC-PTSD-5) for post-traumatic stress [41–43]. PHQ-9 and GAD-7 will be applied if participants screen positive in the respective short versions. The degree of baseline functional disability due to dyspnoea will be measured with the modified Medical Research Council (mMRC) Dyspnoea Scale [44,45]. Secondary outcomes are quality of life and work impact of PCC, assessed using the EuroQol five-dimensional-five-level (EQ-5D-5L) and the Work Productivity and Activity Impairment Questionnaire: General Health V2.0 (WPAI:GH), respectively [46,47]. All these outcomes will be measured at two different time points, nine and twelve months after the positive or negative COVID-19 test.

## Variables

The "Cardiovascular and respiratory events following COVID-19" component of the study will collect information on comorbidities before hospitalisation due to COVID-19; details about the hospitalisation, such as duration, medication administered, nosocomial infections, and need for ventilation; and when available, lifestyle behaviours concerning alcohol use, smoking, and physical exercise. The date of the last contact with the services will be registered, as will the number of days since a positive RT-PCR test for SARS-CoV-2, for each individual to calculate the follow-up time. Data will be retrospectively collected after consultation of the electronic medical record of each patient (indirect data collection) and entered into a fully anonymised specifically for this study. The data dictionary is available in S1A Table. For patients followed in post-COVID consultations, we will collect information from the clinical files regarding PCC presentation and progression over the follow-up time. In addition to the baseline information regarding the patient and infection characteristics, reasons for attending post-COVID consultations, symptoms and clinical progression will be collected.

In the "Physical and mental symptoms following COVID-19" component of the study, interviewers will ask about current and previous comorbidities, sociodemographic information, the level of care required for COVID-19 (home-based, primary care, emergency department, hospital admission, hospital admission in intensive care), and current and previous lifestyle behaviours concerning alcohol use, smoking, and physical exercise. The questionnaire and data dictionary are available in S1B Table.

## Qualitative data collection

The focus of the "Treating and living with Post COVID-19 Condition" component of the study will be the subjectively reported experience of using or working in healthcare and community services for the treatment of PCC symptoms. We will organise three focus groups, each with at least 6 participants–two focus groups with healthcare professionals and one with members of community services–and at least 24 semi-structured interviews with patients with PCC, keeping in mind the eventual need to adjust the number of participants depending on the results that emerge from the qualitative data analysis [48].

In this component, we will employ a semi-structured approach to qualitative data collection. Focus group and interview scripts were designed to support the facilitators' role, with the necessary flexibility. The focus group script covers the following areas: (1) experiences and challenges in providing care to PCC patients; (2) usual trajectories of PCC patients in healthcare and community services; (3) models of care. Participants will be asked to outline the role of PCC in their life before discussing their professional experiences on this matter. For the interviews with PCC patients, a semi-structured interview guide was developed to elicit patients' lived experience with PCC and the role of healthcare and community services in three phases of their experience: (1) pre-COVID and diagnosis; (2) COVID acute phase; (3)

PCC. For each phase, the following broad areas will be explored: illness beliefs; illness trajectory and experiences; impacts in daily life; sources of support; experiences with healthcare and community services; unmet needs and proposals for improving the management of the condition. Interviewees will be invited to outline how PCC affected their life, before engaging in a more focused discussion of their personal experiences with healthcare and community services. Interviews will be conducted preferably in presence or by video-call. All focus groups and interviews will be audio-recorded, transcribed verbatim, and anonymized.

## Analysis

The incidence of each cardiovascular and respiratory events and the combined incidence of at least one of those events will be estimated. Descriptive statistics, using measures of central tendency and dispersion, will be used to characterise the time between COVID-19 diagnosis and the occurrence of a respiratory or cardiovascular event. Cox regression will be used to identify and assess patient and infection factors associated with PCC, defined as the occurrence of at least one of the identified cardiovascular and respiratory events. The analyses will be performed globally, and heterogeneity between hospitals will be accounted for in the models by adding random effects. Hazard ratio (crude and adjusted by sex-age-comorbidities) and the respective 95% confidence intervals will be estimated. Depending on the number of events collected, we will conduct a mediation analysis, considering COVID-19 as a modifier between pre-existing comorbidities and the post-COVID-19 event. Finally, to estimate whether the incidence is above the one expected, we will utilise the background rate identified from the literature to calculate the expected number of events in our study population and compare them with the observed rate. A ratio observed to expected events will be calculated and corresponding 95% confidence intervals will be estimated under a Poisson distribution. Characterisation of patients in PCC consultations will encompass a description of the reason for the consultation using absolute and relative frequencies.

Concerning the "Physical and mental symptoms following COVID-19" component of the study, the prevalence of each symptom at nine and twelve months since notification for exposed and unexposed will be calculated. Absolute and relative frequencies for each symptom and symptom persistence will be measured. Binary logistic regression will be used to identify and assess sociodemographic factors associated with self-reported PCC symptoms at nine and twelve months. Another binary logistic regression will be used to identify and assess sociodemographic factors associated with symptom persistence. Odds ratio (crude and adjusted by sex-age-comorbidities) and the respective 95% confidence intervals will be estimated. Sensitivity analyses will also be performed, excluding case-negative patients diagnosed with COVID-19 following an initial negative test. The quantitative analysis will be performed using R [49].

Data analysis for the "Treating and living with Post Covid-19 Condition" will take place in three stages. During data collection and immediately after each focus group or interview, the facilitators will produce a quick summary, highlighting their first impressions of the most relevant ideas discussed. This will allow for a rapid assessment of the topics raised and the ones that remain underexplored. It will also help guiding the conduct of the following focus groups and interviews, while laying the groundwork for the next stage. In this second analysis stage, the qualitative data from the focus groups and interviews will be submitted to thematic analysis, following an inductive approach, where transcripts will be read and coded to extract themes from the participants' discourses systematically. A computer-assisted qualitative data analysis software (MaxQDA) will be used for data analysis, and intercoder agreement will be sought. In the third stage of analysis, the emerging themes will ground the interpretation and discussion of the narratives of PCC patients and healthcare professionals, which is expected to

contribute to understand the experience and the healthcare needs of these patients when interacting with healthcare and community services. We will use different techniques to strengthen the rigour of the analysis, including reflexivity, triangulation from multiple data sources, and multiple coding.

## Discussion

This article outlines the conceptual framework, rational design and methodological approaches of LOCUS, a multi-component study aiming to broadly characterise the PCC phenomenon.

Despite its strengths, LOCUS also presents some limitations. The first limitation is the manual extraction of information from electronic medical records, which may increase the risk of data collection errors. Furthermore, this limited our ability to include a contemporaneous rather than a historical group. The second limitation is embedded in the non-utilization of ICD coding to get inpatient diagnosis, as this could lead to some vague definitions of outcomes. An exhaustive and specially designed form was created to minimise any possibility of gathering amiss data points. Regarding the second component, the use of questionnaires to assess patient-reported symptoms allows us to have a uniform framework to convene data at nine and 12 months. However, it will also expose the study to recall bias. In order to minimise the impact of the bias on the results, the questionnaire was developed carefully and iteratively, and was tested in two different rounds by members of the research teamAdditionally, we will include individuals tested in 2022, when several restrictive measures were lifted. COVID-19 prevalence was also extremely high, which will difficult the task of recruiting participants with a negative test who never had COVID-19. In the third (qualitative) component, given the novelty of PCC and the potential complexity of a PCC diagnosis, there may be different levels of knowledge and different perceptions about PCC between participants, both among patients and users. While the diversity of opinions is welcome in this component, the focus group and interview guides were developed to accommodate different understandings of what PCC is, focusing on the experiences of participants within the healthcare system, and not so much on their definitions of concepts.

To the best of our knowledge, this is the first multi-component research that studies the PCC phenomena using both qualitative and quantitative methodologies. The results of LOCUS will enlighten the epidemiology and lived experiences of individuals and professionals dealing with PCC. We anticipate the data originated from this study to be used as a stepping stone in innovative COVID-19 policymaking that encompasses the longer-term consequences of this disease as well as COVID-19 management from a clinical and public health perspective.

Several countries may benefit from the data that will arise from this study, as many of those have a similar social, economic, and epidemiological context as Portugal.

## Supporting information

**S1 Table. Data dictionary for the "Cardiovascular and respiratory events following COVID-19" (S1A) and "Physical and mental symptoms following COVID-19" (S1B) components.**
(DOCX)

**S2 Table. Ethics approval of the three components of the study.**
(DOCX)

## Author Contributions

**Conceptualization:** J. P. Dinis Teixeira, Mário J. D. S. Santos, Patrícia Soares, Luísa de Azevedo, Patrícia Barbosa, Andreia Vilas Boas, João V. Cordeiro, Sónia Dias, Marta Fonseca, Ana Rita Goes, Maria João Lobão, Marta Moniz, Sofia Nóbrega, André Peralta-Santos, Víctor Ramos, João Victor Rocha, António Carlos da Silva, Maria da Luz Brazão, Andreia Leite, Carla Nunes.

**Funding acquisition:** J. P. Dinis Teixeira, Mário J. D. S. Santos, Patrícia Soares, Luísa de Azevedo, Patrícia Barbosa, Andreia Vilas Boas, João V. Cordeiro, Sónia Dias, Marta Fonseca, Ana Rita Goes, Maria João Lobão, Marta Moniz, Sofia Nóbrega, André Peralta-Santos, Víctor Ramos, João Victor Rocha, António Carlos da Silva, Maria da Luz Brazão, Andreia Leite, Carla Nunes.

**Methodology:** J. P. Dinis Teixeira, Mário J. D. S. Santos, Patrícia Soares, Andreia Leite.

**Supervision:** Andreia Leite, Carla Nunes.

**Writing – original draft:** J. P. Dinis Teixeira, Mário J. D. S. Santos, Patrícia Soares, João Victor Rocha, Andreia Leite, Carla Nunes.

**Writing – review & editing:** J. P. Dinis Teixeira, Mário J. D. S. Santos, Patrícia Soares, Luísa de Azevedo, Patrícia Barbosa, Andreia Vilas Boas, João V. Cordeiro, Sónia Dias, Marta Fonseca, Ana Rita Goes, Maria João Lobão, Marta Moniz, Sofia Nóbrega, André Peralta-Santos, Víctor Ramos, João Victor Rocha, António Carlos da Silva, Maria da Luz Brazão, Andreia Leite.

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
