## [Decision Letter · Decision Letter 0]

8 Jan 2023

PONE-D-22-31157LOCUS (LOng Covid – Understanding Symptoms, events and use of services in Portugal): A three-component study protocolPLOS ONE

Dear Dr. Dinis Teixeira,

Thank you for submitting your manuscript to PLOS ONE. After careful consideration, we feel that it has merit but does not fully meet PLOS ONE’s publication criteria as it currently stands. Therefore, we invite you to submit a revised version of the manuscript that addresses the points raised during the review process.

We look forward to receiving your revised manuscript.

Kind regards,

Arjun Chandna

Academic Editor

PLOS ONE

Journal Requirements:

2. Please include the full names of the ethics committees that approved your study protocol in the Methods section of your manuscript. Furthermore, please include which ethics committee approved which component of your study in the Methods section of your manuscript.

3. In the ethics statement in the manuscript and in the online submission form, please provide additional information about the patient records/samples that will be used in your retrospective study part. Specifically, please ensure that you have discussed whether all data/samples will be fully anonymized before you accessed them and/or whether the IRB or ethics committee waived the requirement for informed consent. If patients will provide informed written consent to have data/samples from their medical records used in research, please include this information.

4. Please provide additional details regarding participant consent. In the ethics statement in the Methods and online submission information, please ensure that you have specified what type you obtained (for instance, written or verbal, and if verbal, how it was documented and witnessed). If your study included minors, state whether you obtained consent from parents or guardians. If the need for consent was waived by the ethics committee, please include this information.

5. We note that Figure 2 in your submission contain map image which may be copyrighted. All PLOS content is published under the Creative Commons Attribution License (CC BY 4.0), which means that the manuscript, images, and Supporting Information files will be freely available online, and any third party is permitted to access, download, copy, distribute, and use these materials in any way, even commercially, with proper attribution. For these reasons, we cannot publish previously copyrighted maps or satellite images created using proprietary data, such as Google software (Google Maps, Street View, and Earth). For more information, see our copyright guidelines: http://journals.plos.org/plosone/s/licenses-and-copyright.

6. We note that Figure 3 in your submission contain copyrighted images. All PLOS content is published under the Creative Commons Attribution License (CC BY 4.0), which means that the manuscript, images, and Supporting Information files will be freely available online, and any third party is permitted to access, download, copy, distribute, and use these materials in any way, even commercially, with proper attribution. For more information, see our copyright guidelines: http://journals.plos.org/plosone/s/licenses-and-copyright.

a. You may seek permission from the original copyright holder of Figure 3 to publish the content specifically under the CC BY 4.0 license. 

Additional Editor Comments:

Thank you for submitting this manuscript for consideration. Please find below the comments from the reviewers, which require your attention. In particular, please note point 1 from reviewer 2 with regards to the second objective of your study (Physical and Mental symptoms following Covid-19).

In addition to the comments from the reviewers, please clarify:

1. Your definition for symptom persistence in the second objective of your study - is this symptoms present at 9 and 12 months? If so, please justify why these timepoints were chosen, they seem rather arbitrary.

2. Please explain why you are choosing to use the background rate of cardiovascular and respiratory events to compare to those that occur in Covid-19 patients, rather than estimating these rates from Covid-19 negative patients who are hospitalised at the same centres and over the same time period as the Covid-19 positive patients. Given the impact of Covid-19 on health system functioning generally it would seem important to use contemporaneous rather than historical data for the Covid-19 negative patients. This is potentially a limitation that requires acknowledging.

Reviewers' comments:

Reviewer's Responses to Questions

**Comments to the Author**

1. Does the manuscript provide a valid rationale for the proposed study, with clearly identified and justified research questions?

Reviewer #1: Yes

Reviewer #2: Yes

2. Is the protocol technically sound and planned in a manner that will lead to a meaningful outcome and allow testing the stated hypotheses?

Reviewer #1: Yes

Reviewer #2: Yes

3. Is the methodology feasible and described in sufficient detail to allow the work to be replicable?

Reviewer #1: Yes

Reviewer #2: Yes

4. Have the authors described where all data underlying the findings will be made available when the study is complete?

Reviewer #1: No

Reviewer #2: Yes

5. Is the manuscript presented in an intelligible fashion and written in standard English?

Reviewer #1: Yes

Reviewer #2: Yes

6. Review Comments to the Author

You may also provide optional suggestions and comments to authors that they might find helpful in planning their study.

Reviewer #1: This is a well written protocol with clear objectives/components and the procedure. I have few comments for clarification.

Page 7, can you explain who is the data owner who will acquire data from SINAVE?

Providing names and contact number of a person with RT-PCR status bears ethical issues, please add what safeguards have you placed to ensure that the patient’s identification is not disclosed outside the research team. Identification can also lead to potential stigmatization, please add lines on how you will prevent those?

Clarify why you are conducing FGDs? And also add lines on why semi-structured interviews? And why on top of FGDs?

Reviewer #2: My compliments on your study design, as a mixed methods approach to long COVID is essential to furthering our understanding of the condition and involving patients themselves in setting research priorities. I look forward to reading about the results of your research.

I have several comments/suggestions, that you may wish to consider:

1. For the “Physical and mental symptoms following COVID-19 component, you will recruit individuals who tested positive and individuals with a negative test-result. It is unclear to me whether you will match these individuals on age and sex, or even other important confounders such as BMI? If not, why not? This could help in later data analysis.

2. The current state of the pandemic that we are in is vastly different from last year, and I am concerned that the people who are now tested for COVID-19 at an official testing centre are not representative of the general population. How will you go about correcting for this?

3. I am very intrigued by your survival analysis of cardiovascular and respiratory events following COVID-19. How will you determine what proportion of these events are attributable to COVID-19 as opposed to other pre-existing comorbidities? It may be useful to consider COVID-19 as an effect modifier in the relationship between comoborbidities and an event, as this could be of great clinical importance.

4. An important (and generally poorly understood) risk factor for long COVID is immunity resulting from previous vaccination and/or infection. Given that most individuals will have had at least one previous infection and may have been vaccinated multiple times, how do the authors aim to characterise different combinations of this hybrid immunity?

7. PLOS authors have the option to publish the peer review history of their article (what does this mean?). If published, this will include your full peer review and any attached files.

Reviewer #1: No

Reviewer #2: No

---

## [Author Response · Author response to Decision Letter 0]

22 Feb 2023

We appreciate the valuable comments on our manuscript and the recognition of its potential. We have acknowledged the suggestions and updated the manuscript. Please find our answers and indications below for each of the points raised:

We have updated the manuscript according to the Manuscript Body Formatting Guidelines. Headings, figures, tables, supporting information and affiliations are now following the Formatting Guidelines. 

2. Please include the full names of the ethics committees that approved your study protocol in the Methods section of your manuscript. Furthermore, please include which ethics committee approved which component of your study in the Methods section of your manuscript.

The full names of the ethics committees that approved the study protocol were included in Table S2. We have created a subsection “Ethics” linking to the supplementary table, to make this liason clearer, as follows:

“The first component of the study “Cardiovascular and respiratory events following COVID-19” was approved by the Ethics Committee for Health from each participating hospital (Table S2).(…) The second and third parts of the study, “Physical and mental symptoms following COVID-19” and “Treating and living with Post COVID-19 condition” were approved by the Ethics Committee for Health of the Regional Health Administration of Lisbon and Tagus Valley”.

3. In the ethics statement in the manuscript and in the online submission form, please provide additional information about the patient records/samples that will be used in your retrospective study part. Specifically, please ensure that you have discussed whether all data/samples will be fully anonymized before you accessed them and/or whether the IRB or ethics committee waived the requirement for informed consent. If patients will provide informed written consent to have data/samples from their medical records used in research, please include this information.

Thank you for your pertinent comment. All the hospitals waived the requirement for informed consent as all data will be completely anonymised. We added a sentence in the Ethics subsection as follows:

“All hospitals waived the requirement for informed consent, as data will be completely anonymised.”

4. Please provide additional details regarding participant consent. In the ethics statement in the Methods and online submission information, please ensure that you have specified what type you obtained (for instance, written or verbal, and if verbal, how it was documented and witnessed). If your study included minors, state whether you obtained consent from parents or guardians. If the need for consent was waived by the ethics committee, please include this information.

We agree that the protocol was not explicit concerning the informed consent. As mentioned in the previous comment, informed consent was waived for the retrospective study as data will be completely anonymised. We added more information concerning the other two studies in the subsection “Study design”:

“Informed consent will be obtained over the telephone by a trained inquirer. The full consent script will be read to the participant, where the participant can ask questions, refuse or accept to participate in the study. At this stage, the inquirer will have no knowledge of whether the participant had a positive or negative test. Informed consent will be applied by a small number of inquirers (planned to be two), with no further access by the remaining team (apart from the principal investigator), thus limiting the number of team members accessing contacts of individuals. After the informed consent application, and for individuals who consented to participate in the study, the data owner will send us the birthdate and the result of the RT-PCR test for SARS-CoV-2. Inquirers will receive specific training to carry out this task and are bound by deontological professional secrecy rules. Calls may be scheduled at times that are most convenient for the participants.”

Regarding the question related to the informed consent for minors, we apologize for not making it clear in advance, but only adult participants will be enrolled in the study. We have added “below the age of 18 years old” as an exclusion criterion.

5. We note that Figure 2 in your submission contain map image which may be copyrighted. All PLOS content is published under the Creative Commons Attribution License (CC BY 4.0), which means that the manuscript, images, and Supporting Information files will be freely available online, and any third party is permitted to access, download, copy, distribute, and use these materials in any way, even commercially, with proper attribution. For these reasons, we cannot publish previously copyrighted maps or satellite images created using proprietary data, such as Google software (Google Maps, Street View, and Earth). For more information, see our copyright guidelines: http://journals.plos.org/plosone/s/licenses-and-copyright.

Thank you for noticing. To avoid copyright issues, we decided to create a new map in R version 4.2.1.

6. We note that Figure 3 in your submission contain copyrighted images. All PLOS content is published under the Creative Commons Attribution License (CC BY 4.0), which means that the manuscript, images, and Supporting Information files will be freely available online, and any third party is permitted to access, download, copy, distribute, and use these materials in any way, even commercially, with proper attribution. For more information, see our copyright guidelines: http://journals.plos.org/plosone/s/licenses-and-copyright.

Thank you for your comment. We modified figure 3, using different elements. The new version of the Figure contains elements licensed as CC0 Public Domain:

Microscope https://www.publicdomainpictures.net/en/view-image.php?image=360913&picture=dna-test

Person https://publicdomainvectors.org/en/free-clipart/Person-icon-clip-art/88832.html

SARS-CoV-2 https://phil.cdc.gov/details.aspx?pid=23312

Phone https://publicdomainvectors.org/en/free-clipart/Green-phone-icon/70157.html

Captions for the Supporting Information files were added to the end of the manuscript.

Supporting information PLOS.docx was renamed to suit the guidelines. With the same information as the previous submission, two tables were created – S1 Table (with S1A and S1B components) and S2 Table.

Thank you for your comment. All references have been re-checked for any retracted papers and none was found. The list is complete and correct.

In addition to the comments from the reviewers, please clarify:

1. Your definition for symptom persistence in the second objective of your study - is this symptoms present at 9 and 12 months? If so, please justify why these timepoints were chosen, they seem rather arbitrary.

At the time of protocol development several studies already suggested that symptoms could last for at least 6 months, but reports beyond that time were scarce. Thus, we decided to investigate whether symptoms could last for 9 and 12 months. 

2. Please explain why you are choosing to use the background rate of cardiovascular and respiratory events to compare to those that occur in Covid-19 patients, rather than estimating these rates from Covid-19 negative patients who are hospitalised at the same centres and over the same time period as the Covid-19 positive patients. Given the impact of Covid-19 on health system functioning generally it would seem important to use contemporaneous rather than historical data for the Covid-19 negative patients. This is potentially a limitation that requires acknowledging.

Thank you for pointing this out. We appreciate this is a limitation but due to the need to manually extract the information from the records (rather than using a structured databased) we were not able to accommodate a contemporaneous group of negative patients. We thus explicitly added this limitation to the discussion: 

“Furthermore, this limited our ability to include a contemporaneous rather than a historical group.”

Reviewer #1:

Page 7, can you explain who is the data owner who will acquire data from SINAVE?

Thank you for your comment. We have now identified the data owner in the explanation of participants selection.

“The participants selection will be based on medical notifications from the National Epidemiological Surveillance System (SINAVE). Based on data from SINAVE, the data owner (General Directorate of Health - DGS) will provide the research team with information on a list of potential participants in two phases”.

Providing names and contact number of a person with RT-PCR status bears ethical issues, please add what safeguards have you placed to ensure that the patient’s identification is not disclosed outside the research team. Identification can also lead to potential stigmatization, please add lines on how you will prevent those?

We agree this is a sensitive ethical issue. To address these concerns, we designed the application of questionnaires in two stages as explained in the methods section. In the first stage, we will receive a minimal dataset with the variables required to contact participants and additional information will only be received for participants consenting to participate. Following this comment and additional comments from the editor (see above) we have now expanded this section providing further information on application of informed consent and that for the first stage the inquirers team will be reduced so as to limit the number of team members accessing contact information. The full paragraphs now read:

“In the first phase, the data owner will provide the names and contact numbers (landline or mobile) of individuals who had an RT-PCR test nine months before the beginning of data collection. Informed consent will be obtained over the telephone by a trained inquirer. The full consent script will be read to the participant, where the participant can ask questions, refuse or accept to participate in the study. At this stage, the inquirer will have no knowledge of whether the participant had a positive or negative test. Informed consent will be applied by a small number of inquirers (planned to be two), with no further access by the remaining team (apart from the principal investigator), thus limiting the number of team members accessing contacts of individuals. After the informed consent application, and for individuals who consented to participate in the study, the data owner will send us the birthdate and the result of the RT-PCR test for SARS-CoV-2. Inquirers will receive specific training to carry out this task and are bound by deontological professional secrecy rules. Calls may be scheduled at times that are most convenient for the participants.”

Clarify why you are conducing FGDs? And also add lines on why semi-structured interviews? And why on top of FGDs?

Focus group discussions with healthcare professionals were considered more appropriate to collect data with this group given that group processes help people identify and clarify their views, because participants are encouraged to share, confront and comment their views. Therefore, focus groups often bring to the discussion experiences and perspectives that might not be voiced in individual interviews, enriching the data. Besides, focus groups bring the opportunity to develop shared perspectives, which was considered particularly valuable for understanding overall needs and resources and specificities from particular settings as well as for the articulation of proposals regarding models of care.

In the case of PCC patients, we considered that a group process could undermine the collection of the detailed experience of participants, missing important idiosyncrasies. In addition, the specific burden brought by PCC could prevent the participation of some patients in a group process, which duration and rhythm may be hard to adjust to the specific needs and circumstances of each participant.

Reviewer #2: 

1. For the ”Physical and mental symptoms following COVID-19" component, you will recruit individuals who tested positive and individuals with a negative test-result. It is unclear to me whether you will match these individuals on age and sex, or even other important confounders such as BMI? If not, why not? This could help in later data analysis.

Thank you for your relevant comment. We agree that it would be interesting to match negative and positive cases based on some characteristics, such as age and gender. However, given the high prevalence of COVID-19 in Portugal, it will be difficult to recruit individuals who never had a positive COVID-19 test. Thus, we will not match positive and negative test-result to improve our chance of reaching 500 individuals with a negative test who accept to participate in the study. 

2. The current state of the pandemic that we are in is vastly different from last year, and I am concerned that the people who are now tested for COVID-19 at an official testing centre are not representative of the general population. How will you go about correcting for this?

Thank you for your comment. We agree that testing changes depending on the time of the pandemic. However, we are already discussing data transfer with DGS and asked for cases tested in June 2022, when the incidence was extremely high in Portugal. Thus, we believe that this issue will be mitigated. Nevertheless, we acknowledge that most restrictive measures were already lifted and that the proportion of individuals who tested negative for the first time might be more difficult at the end of the pandemic. We added the following sentence to the discussion:

“(…) Additionally, we will include individuals tested in 2022, when several restrictive measures were lifted. COVID-19 prevalence was also extremely high, which will difficult the task of recruiting participants with a negative test who never had COVID-19. (…)”

3. I am very intrigued by your survival analysis of cardiovascular and respiratory events following COVID-19. How will you determine what proportion of these events are attributable to COVID-19 as opposed to other pre-existing comorbidities? It may be useful to consider COVID-19 as an effect modifier in the relationship between comorbidities and an event, as this could be of great clinical importance.

We agree with your comment. Some cardiovascular and respiratory events might occur due to pre-existing comorbidities. Thus, we will adjust the analysis for these pre-existing comorbidities. In case we have enough cardiovascular or respiratory events to analyse individually, we will consider a mediation analysis for that specific event. We added a clarification to the “Analysis” subsection: 

“Depending on the number of events collected, we will conduct a mediation analysis, considering COVID-19 as a modifier between pre-existing comorbidities and the post-COVID-19 event.”

4. An important (and generally poorly understood) risk factor for long COVID is immunity resulting from previous vaccination and/or infection. Given that most individuals will have had at least one previous infection and may have been vaccinated multiple times, how do the authors aim to characterise different combinations of this hybrid immunity?

Thank you for this important comment. We agree that this is an important issue, and it would be extremely relevant to have further insights. While we plan to characterise the diversity of different combinations, we anticipate a high number of possible combinations making it difficult to assess all of them.

---

## [Decision Letter · Decision Letter 1]

14 Apr 2023

LOCUS (LOng Covid – Understanding Symptoms, events and use of services in Portugal): A three-component study protocol

PONE-D-22-31157R1

Dear Dr Jose Teixeira,

We’re pleased to inform you that your manuscript has been judged scientifically suitable for publication and will be formally accepted for publication once it meets all outstanding technical requirements.

Kind regards,

Dirceu Henrique Paulo Mabunda, M.D.

Academic Editor

PLOS ONE

Additional Editor Comments (optional):

Congratulations you did all reviewers comments.

Reviewers' comments:

Reviewer 1

This is a well written protocol with clear objectives/components and the procedure. I have few comments for clarification.

Page 7, can you explain who is the data owner who will acquire data from SINAVE?

Providing names and contact number of a person with RT-PCR status bears ethical issues, please add what safeguards have you placed to ensure that the patient’s identification is not disclosed outside the research team. Identification can also lead to potential stigmatization, please add lines on how you will prevent those?

Clarify why you are conducing FGDs? And also add lines on why semi-structured interviews? And why on top of FGDs?

Reviewer 2 

My compliments on your study design, as a mixed methods approach to long COVID is essential to furthering our understanding of the condition and involving patients themselves in setting research priorities. I look forward to reading about the results of your research.

I have several comments/suggestions, that you may wish to consider:

1. For the “Physical and mental symptoms following COVID-19 component, you will recruit individuals who tested positive and individuals with a negative test-result. It is unclear to me whether you will match these individuals on age and sex, or even other important confounders such as BMI? If not, why not? This could help in later data analysis.

2. The current state of the pandemic that we are in is vastly different from last year, and I am concerned that the people who are now tested for COVID-19 at an official testing centre are not representative of the general population. How will you go about correcting for this? 

3. I am very intrigued by your survival analysis of cardiovascular and respiratory events following COVID-19. How will you determine what proportion of these events are attributable to COVID-19 as opposed to other pre-existing comorbidities? It may be useful to consider COVID-19 as an effect modifier in the relationship between comoborbidities and an event, as this could be of great clinical importance.

4. An important (and generally poorly understood) risk factor for long COVID is immunity resulting from previous vaccination and/or infection. Given that most individuals will have had at least one previous infection and may have been vaccinated multiple times, how do the authors aim to characterise different combinations of this hybrid immunity?

Reviewer's Responses to Questions

**Comments to the Author**

1. Does the manuscript provide a valid rationale for the proposed study, with clearly identified and justified research questions?

Reviewer #2: Yes

Reviewer #3: Yes

2. Is the protocol technically sound and planned in a manner that will lead to a meaningful outcome and allow testing the stated hypotheses?

Reviewer #2: Yes

Reviewer #3: Yes

3. Is the methodology feasible and described in sufficient detail to allow the work to be replicable?

Reviewer #2: Yes

Reviewer #3: Yes

4. Have the authors described where all data underlying the findings will be made available when the study is complete?

Reviewer #2: Yes

Reviewer #3: Yes

5. Is the manuscript presented in an intelligible fashion and written in standard English?

Reviewer #2: Yes

Reviewer #3: Yes

6. Review Comments to the Author

You may also provide optional suggestions and comments to authors that they might find helpful in planning their study.

Reviewer #2: Thank you for responding to my previous comments.

In your response to my second point, you discuss the difficulty in case-finding individuals with a negative test. I have two main concerns about your response.

Firstly, even during high-transmission/high-prevalence periods of time, the majority of individuals with respiratory symptoms who are tested will not have a SARS-CoV-2 infection, therefore the statement that finding negative cases will be difficult, does not make sense. I suggest that you reword the statement.

Second, my initial comment was actually concerning the sampling bias introduced due to changing regulations for testing, not about recruitment. Following the availability of self-tests, for instance, not everyone will continue to visit a formal testing centre to receive their diagnosis. My question therefore was whether the authors take into account that the type of person to be tested (your sampling frame) may change over time. A supplementary table comparing baseline characteristics of individuals tested (negative or positive) during difference phases of the pandemic could be very insightful here.

Reviewer #3: The authors have addressed most of the reviewers comments. Where this was not possible, the authors acknowledge the limitations. It is a strength of this project to bring together three studies that address different questions around PCC. This is very valuable for increasing the knowledge base.

The only comment I would make at this point is the collaboration and communication to health care professionals, representatves of persons living with PCC and health authorities. It would be good to have a plan for science to public and to policy makers communication so that the results of this study contribute to the development of a common understand of PCC and how to best support persons living with PCC.

7. PLOS authors have the option to publish the peer review history of their article (what does this mean?). If published, this will include your full peer review and any attached files.

Reviewer #2: No

Reviewer #3: **Yes: **Milo Puhan

---

## [Editor Report · Acceptance letter]

18 Apr 2023

PONE-D-22-31157R1 

­­­­LOCUS (LOng Covid – Understanding Symptoms, events and use of services in Portugal): A three-component study protocol 

Dear Dr. Dinis Teixeira:

I'm pleased to inform you that your manuscript has been deemed suitable for publication in PLOS ONE. Congratulations! Your manuscript is now with our production department. 

Kind regards, 

on behalf of

Dr. Dirceu Henrique Paulo Mabunda 

Academic Editor

PLOS ONE